# On Modelling Parasitic Solidification Due to Heat Loss at Submerged Entry Nozzle Region of Continuous Casting Mold

Alexander Vakhrushev [1], Abdellah Kharicha [1,*], Menghuai Wu [2], Andreas Ludwig [2], Yong Tang [3], Gernot Hackl [3], Gerald Nitzl [4], Josef Watzinger [5] and Jan Bohacek [6]

1 Christian-Doppler Lab for Metallurgical Applications of Magnetohydrodynamics, Montanuniversität Leoben, 8700 Leoben, Austria; alexander.vakhrushev@unileoben.ac.at
2 Chair of Simulation and Modeling of Metallurgical Processes, Department of Metallurgy, Montanuniversität Leoben, 8700 Leoben, Austria; menghuai.wu@unileoben.ac.at (M.W.); andreas.ludwig@unileoben.ac.at (A.L.)
3 RHI Magnesita GmbH, 8700 Leoben, Austria; yong.tang@RHIMagnesita.com (Y.T.); gernot.hackl@RHIMagnesita.com (G.H.)
4 RHI Magnesita GmbH, 1120 Vienna, Austria; gerald.nitzl@RHIMagnesita.com
5 Primetals Technologies, 4031 Linz, Austria; josef.watzinger@primetals.com
6 Heat Transfer and Fluid Flow Laboratory, Faculty of Mechanical Engineering, Brno University of Technology, 61669 Brno, Czech Republic; Jan.Bohacek@vut.cz
* Correspondence: abdellah.kharicha@unileoben.ac.at

**Abstract:** Continuous casting (CC) is one of the most important processes of steel production; it features a high production rate and close to the net shape. The quality improvement of final CC products is an important goal of scientific research. One of the defining issues of this goal is the stability of the casting process. The clogging of submerged entry nozzles (SENs) typically results in asymmetric mold flow, uneven solidification, meniscus fluctuations, and possible slag entrapment. Analyses of retained SENs have evidenced the solidification of entrapped melt inside clog material. The experimental study of these phenomena has significant difficulties that make numerical simulation a perfect investigation tool. In the present study, verified 2D simulations were performed with an advanced multi-material model based on a newly presented single mesh approach for the liquid and solid regions. Implemented as an in-house code using the OpenFOAM finite volume method libraries, it aggregated the liquid melt flow, solidification of the steel, and heat transfer through the refractory SENs, copper mold plates, and the slag layer, including its convection. The introduced novel technique dynamically couples the momentum at the steel/slag interface without complex multi-phase interface tracking. The following scenarios were studied: (i) SEN with proper fiber insulation, (ii) partial damage of SEN insulation, and (iii) complete damage of SEN insulation. A uniform 12 mm clog layer with 45% entrapped liquid steel was additionally considered. The simulations showed that parasitic solidification occurred inside an SEN bore with partially or completely absent insulation. SEN clogging was found to promote the solidification of the entrapped melt; without SEN insulation, it could overgrow the clogged region. The jet flow was shown to be accelerated due to the combined effect of the clogging and parasitic solidification; simultaneously, the superheat transport was impaired inside the mold cavity.

**Keywords:** heat and mass transfer; solidification; clogging; submerged entry nozzle; continuous casting; OpenFOAM®

## 1. Introduction

Continuous casting (CC) is the process whereby molten steel is solidified into a semi-finished billet, bloom, or slab for subsequent forming in rolling mills. Prior to the introduction of CC in the 1950s, steel was poured into stationary molds to form so-called ingots. Since then, continuous casting has evolved to achieve improved quality and productivity while aiming to increase the cost-efficiency. Moreover, some technologies

in the metallurgical industry (e.g., ingot casting and electro slag remelting) have been further developed and transformed into continuous or semi-continuous processes. The main advantages are high production rates and near net manufacturing shapes.

In Figure 1a, a typical continuous caster configuration is displayed. Steel is tapped from an electric or basic oxygen furnace into a ladle, from which is taken to a continuous casting machine. The ladle is raised onto a turret that rotates the ladle into the casting position above the tundish. A special tundish design was developed to stabilize a flow and to initially remove the non-metallic inclusions. However, small fractions of the non-metallic inclusion are drained into a mold [1]. Liquid steel is fed into a water-cooled copper mold through a submerged entry nozzle (SEN). Solidification begins in the mold (at the so-called primary cooling zone) and continues through the secondary cooling zone, where the semi-solid strand is supported and guided by the rolls, which are required to maintain ferrostatic pressure/product shape through final solidification. The strand is typically straightened, torch-cut, and discharged for intermediate storage or hot-charged for finished rolling.

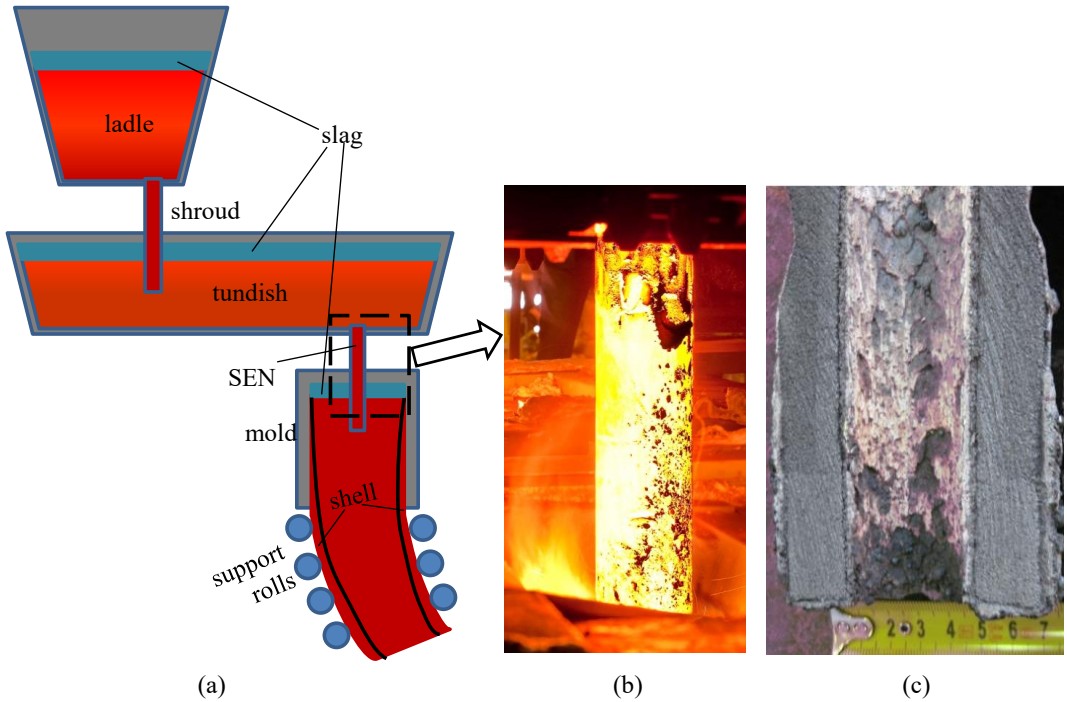

(a)          (b)          (c)

**Figure 1.** Continuous casting: (**a**) process schematics; (**b**) a submerged entry nozzle in operation; (**c**) a central cut of an SEN after 10 heats.

Well-known issues that arise due to the specific features of CC include surface or internal defect formation due to the high cooling and rapid solidification, the clogging of SENs, and the possibility of breakouts. The adjustment of the real CC process is expensive and requires time. Mathematical models and experimental work [2,3] related to solidification phenomena and the turbulent flow/two phase region (the so-called mushy zone) interaction [4] have been intensively developed, enhanced, and applied for the last 30 years [5–7] in order to investigate and improve the casting process [8–13].

Shell-related phenomena originate in the CC mold and the secondary cooling zones. Some counter-measures such as mold oscillation are necessary to avoid shell tearing, sticking, and breakouts [14,15]. The influence of casting speed and mold oscillation on the slag consumption was discussed by Jonayat and Thomas [16]. Mathematical modelling aids in proper design and operating practices [17–20]. Several numerical models have been developed for the contact problem and air gap formation [20–23]. Deformation

models coupled with the fluid flow, heat transfer, and solidification have been described elsewhere [24–29].

Efficient water-spraying in the secondary cooling zone is an important CC topic. Recently, Hnizdil et al. [30] worked out an improved prediction of the Leidenfrost temperature. Kotrbacek et al. [31] combined experimental investigations to adjust the classical estimation of the HTC by introducing a correlation function [31]. A novel method to flatten the cooling jet with the nozzles and deflecting vanes was presented by Bohacek et al. [32].

Some of the initial shell formation issues can be impaired by the EMBr, as recently reviewed in [33–36]. The EMS can be a useful tool to improve a disturbed flow pattern due to SEN clogging [37]. It was shown that the closure of the induced e-current through the shell [38,39] alternates the topology of the Lorentz force from damping to acceleration and leads to reverse flow formation [40]. Therefore, it is essential to correctly predict solid shell growth while considering the full heat balance in an SEN/mold/slag system, which was achieved for and is detailed in the present paper.

Free surface behavior, slag/gas bubble entrapment issue, etc., are of immense importance [41–43]. Both experimental work and advanced model development have been the topics of recent research [44–48]. Lopez et al. [49] addressed slag infiltration, interfacial resistance, and a lubrication index in aiming to re-formulate outdated quality concepts.

Aiming for efficient energy costs is crucial at each stage of continuous casting. Moreover, the energy losses in the marked area in Figure 1a could lead to the parasitic solidification inside an SEN bore, thus causing the failure of the whole casting process.

Flow control products such as a ladle shroud (LS) and SEN are used to protect the transported molten steel from re-oxidation [50]. The complexity of SEN design [51] defines the mold flow pattern, meniscus calmness, the melting rate of flux slag, etc.

Refractory properties are important for the heat transfer, wear, and thermal stress resistance of a material [52,53]; gas bubble formation [54]; and clogging rate, among others. SEN clogging causes an asymmetric flow pattern, parasitic solidification at an SEN, hook formation at a meniscus, local shell thinning, and breakouts since the forced convection plays a crucial role in solidified shell formation [55,56].

Practical observations during CC operation (Figure 1b) for SEN clogging are shown in Figure 1c, where it can be seen that the inner surface is covered by a clog layer, so solidified steel is detected in the clogging material. The strongly radiating surface of the SEN in Figure 1b indicates significant heat losses. Multi-parameter analysis, describing clog shape, was performed by Hua et al. [57] to investigate block rate's influence on the flow pattern in the mold and on steel–slag interface fluctuations. Gutiérrez et al. [58] studied three clog counter-measures such as a flow control, proper raw materials, and the reduction of an SEN;s internal roughness. The application of new materials and technologies, anti-clog alloy treatment, and the typical composition of clogging materials have been presented elsewhere [50,59,60].

Recently, an advanced model was developed based on turbulent quantities near the refractory wall [61,62] to study whether clogging or parasitic solidification dominates. This model was employed together with a solidification model to analyze which phenomenon is dominant [63,64]; using an experimental setup with melt velocities of ~4–5 m/s, solidification inside the SEN was found to follow the clogging front.

However, most of these topics require full CFD modelling, which is extremely computationally expensive. Moreover, focus has been limited to local regions of interest. To speed-up the coupled problem simulation, reduced order models have been continuously developed [23,65,66]. The recently proposed recurrence methodology (rCFD) reported in [67] uses collected statistics to extrapolate simulation results for the steady-state processes and (partially) the transient processes [68–72]. Efficient techniques to solve the linear system of equations for the CFD problems using graphical cards have become very popular [73]. Koric and Abueidda demonstrated that deep learning methods, trained by simulation results, are able to recover temperature histories [74].

A legitimate simplification is dimension reduction: a 3D problem is solved using 2D or 1D approximations [21,23,65]. A 2D multiphase model was used by Jonayat and Thomas to model thermal behavior and slag consumption in the meniscus region during an oscillation cycle [16]. Recently, a combined 3D and 2D hybrid model, verified by an experiment, was applied to simulate flow, solidification, and centerline segregation during slab CC [75,76]. Mitchell and Vynnycky conducted a detailed study that showed that by using asymptotic methods, key CC quantities can be recovered using a simple heat transfer model that considers turbulence and thermo-mechanics [77]. Newly developed on-line monitoring systems can use an advanced post-processing techniques as well [78].

In the present study, the authors kept the complex multiphase phenomena during continuous casting. The melt and slag flows were modeled together with the heat transfer in a copper mold and refractory SEN. The solidification was considered both in the slab region and the vicinity of the inner refractory wall. A 2D single mesh simulation was performed to avoid modelling the free surface tracking and conjugated heat transfer between solid and liquid regions. The validity of 2D approximation for CC process modelling has been widely proven in the literature.

A linear system of discretized PDEs, describing fluid flow, heat transfer, and solidification, was obtained using the finite volume method (FVM) in the frames of the open-source CFD package OpenFOAM®. Our newly presented approach manipulates the implicit coefficients and the source terms of a discretized linear system of equations on the basic level to separately treat solid and liquid regions. The interface between the melt and slag layer is assumed to be planar, so no multi-phase interface tracking is required. A novel technique to dynamically couple the momentum at the liquid interface was developed and applied. The approach appears to be robust and computationally efficient.

Additionally, the authors of the present study considered the formation of clogging inside an SEN. A uniform 12 mm clog layer with 45% entrapped liquid steel was assumed to be non-permeable for the melt flow. The process parameters and the clogging material properties were taken from the literature [50,64].

Based on the developed model, the heat transfer in the SEN region was considered for different operational modes: (i) SEN with proper fiber insulation; (ii) the partial damage or absence of a fiber layer; and (iii) the additional formation of a 12 mm SEN clog. For these scenarios, the heat losses through the refractory were modeled and the conditions for the parasitic solidification were considered. The melt flow and superheat distribution in the CC mold under these conditions were analyzed.

## 2. Materials and Methods

The schematics of the simulated continuous casting process and the modelling domain, as shown in Figure 2, consist of several regions: a liquid melt pool where solidification can occur; a liquid slag layer; a water-cooled CC mold with an insulating slag skin layer; a refractory SEN with/without fiber insulation (outer wall) and a clog layer (inner wall).

The corresponding set of the mass, momentum, and energy conservation equations are defined in this section. It should be mentioned that a single Eulerian mesh was used with the aim to perform fast calculations of the presented multiphase phenomenon. A novel numerical approach is disclosed in this section. Summarized settings of the simulated cases and their parameters are listed in the last subsection.

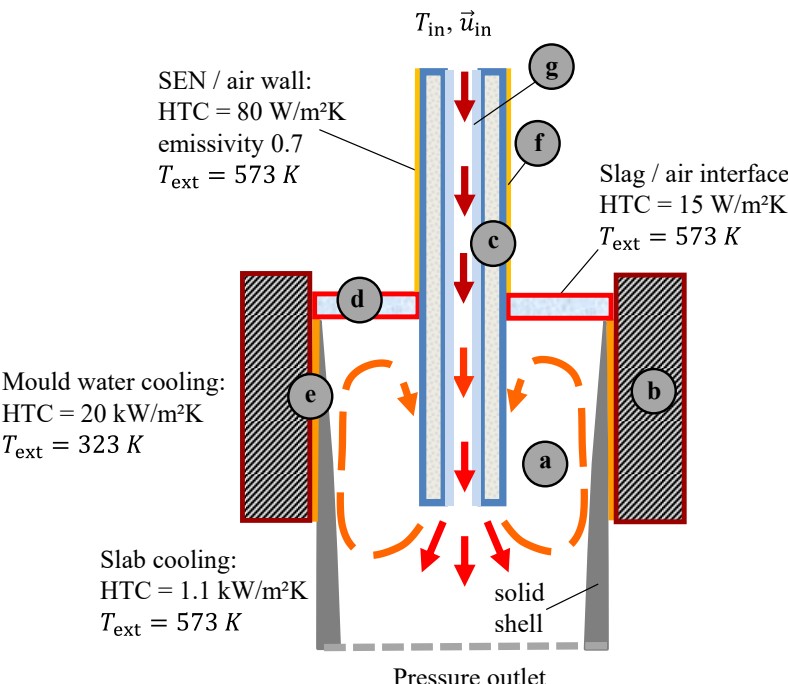

**Figure 2.** Schematics of the simulation domain: (**a**) melt pool, (**b**) water-cooled CC mold, (**c**) refractory SEN wall, (**d**) slag band, (**e**) 2 mm slag skin, (**f**) fiber insulation, and (**g**) SEN clogging layer.

### 2.1. Modelling Liquid Flow

In the liquid melt/slag region, continuity and the momentum equations (in incompressible formulation) are combined with the energy equation. The first two equations are written for the liquid melt velocity $\vec{u}$ as

$$\nabla \cdot \vec{u} = 0 \tag{1}$$

$$\rho \left( \frac{\partial \vec{u}}{\partial t} + \nabla \cdot \left( \vec{u} \otimes \vec{u} \right) \right) = \mu \, \Delta \, \vec{u} - \nabla p + \vec{S}_{\text{Darcy}} \tag{2}$$

where $\rho$ and $\mu$ are the melt density and dynamic viscosity, respectively; $p$ represents a pressure field; and $\vec{S}_{\text{Darcy}}$ is a momentum sink term inside a two phase region (so-called mushy zone) that is described later. The turbulence was neglected in the current study, partially due to the 2D formulation, and will be a topic of future study.

### 2.2. Solidification

The enthalpy-based mixture solidification model [5–7] was applied to resolve the formation of the solidified shell in the mold/strand region and possible parasitic solidification along the inner wall of an SEN. For a detailed description of the model, the reader is referred to the work of Vakhrushev et al. [79]. Briefly, a mixture approach that combines a liquid $\ell$-phase and a solid $s$-phase is used. These phases are quantified by their volume fractions $f_\ell + f_s = 1$. The energy equation for the temperature $T$ can be written as follows:

$$\rho C_p \left( \frac{\partial T}{\partial t} + \nabla \cdot \left( \vec{u} T \right) \right) = \lambda \, \Delta T + S_e, \tag{3}$$

where $\lambda$ is the thermal conductivity and the latent heat term $S_e$ is calculated as

$$S_e = \rho L \left( \frac{\partial f_s}{\partial t} + \vec{u}_{\text{pull}} \cdot \nabla f_s \right), \tag{4}$$

With the latent heat of fusion $L$ and casting speed $\vec{u}_{\text{pull}}$.

The flow in the mushy zone is determined according to the Darcy's permeability, which evolution is determined by the temperature according to a $f_s - T$ relation [80,81]:

$$S_{\text{Darcy}} = \frac{\mu}{6 \times 10^{-4} \lambda_1^2} \frac{f_s^2}{f_\ell^3} (\vec{u} - \vec{u}_{\text{ref}}), \tag{5}$$

where $\lambda_1$ is the dendrite primary arm spacing.

For the slag region, the heat transfer is reduced to the advection–diffusion problem to calculate the temperature field. The solidification and remelting of the slag were left out in this study, though they are interesting and important topics for the future. In the solid body regions (CC mold and refractory), a single conduction problem is solved:

$$\rho C_p \frac{\partial T}{\partial t} = \lambda \, \Delta \, T, \tag{6}$$

where $C_p$ represents the specific heat of the corresponding materials. The material properties are summarized in Table 1.

**Table 1.** Material properties.

| Component Material | Density $\rho$ (kg·m$^{-3}$) | Specific Heat $C_p$ (J·kg$^{-1}$·K$^{-1}$) | Thermal Conductivity $\lambda$ (W·m$^{-1}$·K$^{-1}$) | Thermal Diffusivity $\alpha$ (m$^{-2}$·s$^{-1}$) | Dynamic Viscosity $\mu$ (kg·m$^{-1}$·s$^{-1}$) |
|---|---|---|---|---|---|
| Melt [81] | 7020 | 700 | 36.4 | $7.41 \times 10^{-6}$ | 0.006 |
| Copper mold | 8980 | 380 | 390.0 | $1.14 \times 10^{-4}$ | |
| Refractory [64] | 2430 | 1416 | 18.0 | $5.23 \times 10^{-6}$ | |
| Liquid slag | 2700 | 1250 | 4.0 | $1.11 \times 10^{-6}$ | 0.002 |
| Slag skin | 3000 | 1000 | 0.5 | $1.67 \times 10^{-7}$ | |
| Clog [64] | 3700 | 700 | 35.0 | $1.35 \times 10^{-5}$ | |

### 2.3. Single Mesh Approach

One of the key features of the present work is the introduction of a single mesh approach. One set of continuity, momentum, and energy equations was solved for the multi-material region.

A semi-discrete form of the momentum Equation (2) using the finite-volume method (FVM) can be expressed for the velocity vector components in the cell centers P based on the neighboring cell values N and the pressure gradient as follows:

$$\mathbf{A} \cdot u_{\text{P}}^i = \vec{H}(u_{\text{N}}^i) - \frac{\partial}{\partial i}(p), \quad i = \{x, y\}. \tag{7}$$

The implicit coefficients of the linear system of equations $\mathbf{A}$, the right-hand side (RHS) operator $\vec{H}$, and pressure gradient $\frac{\partial}{\partial i}(p)$ depend on the FVM discretization schemes used in the simulation. For the solution of the linear system of Equation (7), the following constrains were applied:

- It was not modified in the melt region (Figure 2a); for the rest of domain, the Darcy drag term $\vec{S}_{\text{Darcy}}$ was excluded.
- In the solid body regions including the CC mold and the refractory SEN (Figure 2b,c), the velocities in the Equation (7) were explicitly set to zero by corresponding manipulations with the matrix coefficients.
- An interface between the melt and the liquid slag (Figure 2d), where the momentum transfer was only coupled in the horizontal direction, was introduced. The implementation of an algorithm is disclosed in the next section.

For the closure of the mixture model description, Equations (3) and (4) (which include the solidification terms of the latent heat release due to the phase change and its advection by the dendrites network) were included, as described in [79]. The latent heat release $S_e$ was only allowed in the liquid melt. The reference velocity $\vec{u}_{\text{ref}}$ in the Darcy term in Equation (5) was set to the casting velocity $\vec{u}_{\text{pull}}$ in the region close to the CC mold and the strand hot surface, and it was set to zero elsewhere. Thus, the solid shell was constantly withdrawn in the mold/strand region. On the contrary, the mush statically grew close to the SEN wall if parasitic solidification occurred.

The advection term was automatically removed in the refractory region, limiting it to the simple conduction problem exactly in the same manner as in the mold domain.

*2.4. Momentum Coupling at the Melt/Slag Interface*

The fluid flow (see Equations (1) and (2)) was solved both for the slag and melt on a single finite-volume mesh. Here, an assumption of a constant interface position between the slag and melt is proposed (see Figure 3): the interface forces and the interface waiving are ignored, but the heat and momentum transfer between these two phases are fully considered. Following the chart in Figure 3, the corresponding procedure was performed for all cell interfaces with *i*-index on the melt side and the *j*-index for the slag:

1. The previous iteration velocity values $\vec{u}_i = (u_x;\ u_y)^{\text{T}}$ were taken from the melt-side interface cells.
2. They were stored for the corresponding pairs of the neighboring slag cells with the inverted vertical component, such as $\vec{u}_j^* = (u_x;\ -u_y)^{\text{T}}$. The momentum flux, normal to the melt/slag interface, automatically became zero through cell-center to face-center interpolation. This approach, to some extent, reflects the ghost cell method. Simultaneously, the linear momentum, parallel to the interface, was transferred into the slag bulk due to the viscous shear stress, and recirculation zones were established.

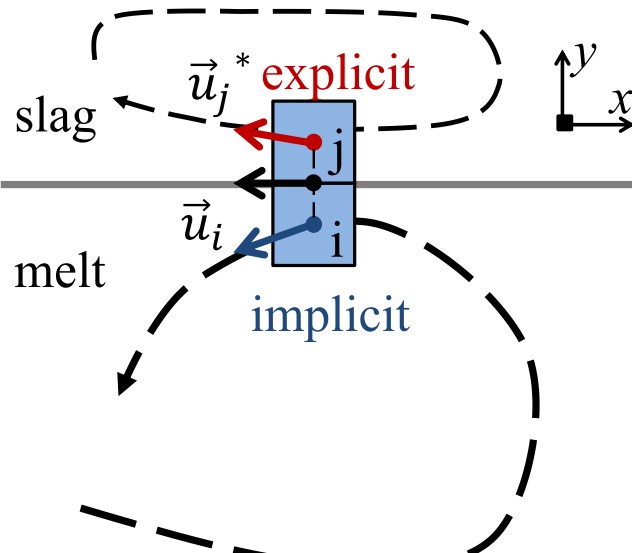

**Figure 3.** Splitting convection at the melt/slag interface and the coupling momentum transfer of two phases.

To keep a solution procedure of the momentum equation consistent and converging, the following manipulations are performed with the linear system matrix in Equation (7):

1. For all off-diagonal elements (neighboring cells), the $\vec{u}^*$ values were multiplied by corresponding coefficients and moved to the source part in RHS.

2. The diagonal and source coefficients for the slag interface cells were set to 1 and $\vec{u}^*$, respectively; therefore, the linear solver naturally produced aimed values $\vec{u}^*$. Off-diagonal coefficients were set to zero.

3. After solution of the linear system (7), the melt interface values $\vec{u}$ were updated and did not match the corresponding $\vec{u}^*$; therefore, an iterative procedure was repeated to obtain the full convergence of the explicit and implicit parts of algorithm in Figure 3.

This single-mesh approach is efficient because no extra multi-region coupling or free surface tracking is required. The developed numerical model included liquid melt flow; the solidification of the steel; heat transfer in the water-cooled CC mold, through the slag skin, and in the slag layer (including its convection); and heat losses to the air from different parts of the system. The simulation was performed in 2D. The validity of this approach is discussed in the introduction. The surface temperatures of the SEN were verified based on the experimental measurements by RHI Magnesita GmbH [50].

### 2.5. Material Properties and Case Setup

The material properties are gathered in Table 1. For the solidification modelling, a low-alloy steel (C 0.05 wt%) was selected. The physical properties were calculated using the IDS software [81] at a typical CC cooling rate of 10 K/s to compare the obtained results with previous studies [50,64]. The casting parameters were adjusted to combine the settings from both publications. As a result, an alloy was cast in the simulation with a casting speed of 2.4 m/min, which resulted in inlet velocities of ~1 m/s. The simulated process parameters are summarized in Table 2.

**Table 2.** Simulated process parameters.

| Parameter | Value |
|---|---|
| $T_{\text{solidus}}$, K | 1723 |
| $T_{\text{liquidus}}$, K | 1738 |
| $T_{\text{in}}$, K | 1753 |
| $\vec{u}_{\text{pull}}$, m·min$^{-1}$ | 2.4 |
| Mold width, mm | 1500 |
| Copper plate, mm | 40 |
| SEN inner/outer radius, mm | 36/62 |
| SEN immerse depth, mm | 200 |
| Clog layer thickness, mm | 0 or 12 |

Three different scenarios were included in the present work: (i) solidification in an SEN insulated with a fiber layer [50]; (ii) low insulation of the SEN (possibly damaged), with a 3-times reduction of its thermal resistance; and (iii) no fiber insulation layer. All cases were supplemented with a second option when the SEN was partially blocked with a clog layer (see Tables 1 and 2 for properties). Ultimately, 6 calculation cases were considered.

In the current study, the clog layer was assumed to be uniform with a moderate thickness of 12 mm [64]. The clogging was assumed to be non-permeable for the flow. It contained of the 45% volume fraction of the melt that could solidify [64]. This simplified model provided a general impression regarding clog influence on the parasitic solidification inside the SEN.

The distribution of the thermal properties of the system components is shown in Figure 4, where thermal diffusivity is used as a measure of the heat transfer inside the materials. As can be seen, the melt and refractory thermal properties were very close to each other, so a significant heat loss through the refractory could be expected.

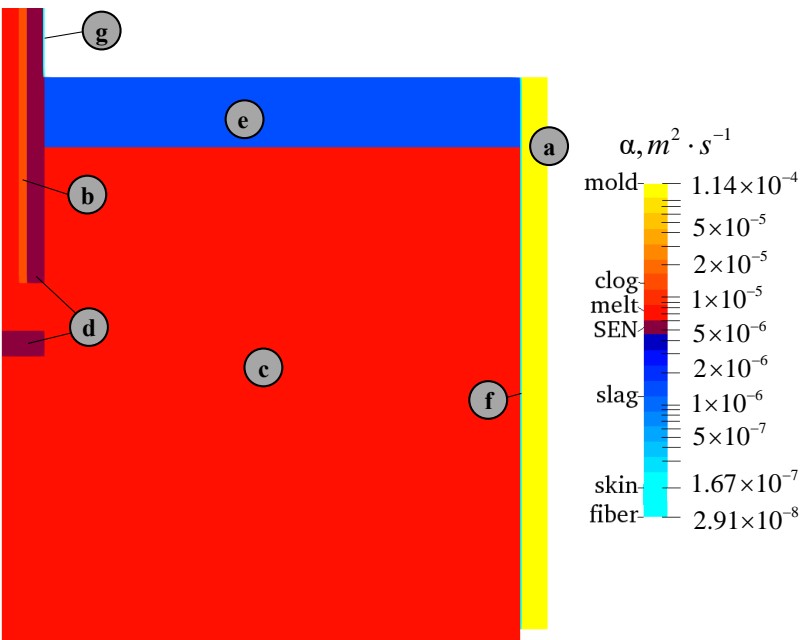

**Figure 4.** Thermal diffusivity $\alpha$ of the casting system components (in descending order): (**a**) mold, (**b**) clogging material, (**c**) liquid melt, (**d**) refractory, (**e**) liquid slag, (**f**) slag skin, and (**g**) SEN fiber insulation.

### 2.6. Numerical Model Performance

The governing equations of the solidification model [79] were implemented by the authors with the in-house source code using enthalpy–porosity approach. The finite volume method library from the OpenFOAM® CFD software package [82] was utilized to discretize the initial physical problem in the tensor-vector form and solved using high-performance calculations system.

The simulation used a 2D mesh consisting of 75,000 computational cells. Second order temporal and spatial discretization was applied. All simulations were performed to get a quasi-steady state solution to collect reliable time-averaged statistics for the velocity and temperature fields to perform the further analysis of the results. Therefore, each simulation case ran for 3600 physical seconds (1 h of the CC process) using parallel calculations on the 6-core node of an Intel Xeon E5-1650 v4 @ 3.60 GHz architecture with 64 GB of RAM. This simulation required 288 s (less than 5 min) of the wall-clock time. Considering that a coupled flow inside the mold cavity (including solidification) and the slag layer was resolved together with the heat transfer in the mold and refractory, the newly presented numerical method is highly efficient.

## 3. Results and Discussion

The simulation setup consisted of the multi-area domain (see Figure 2): a hot liquid metal was fed via an SEN into the continuous casting mold. The top surface of the liquid metal was covered with a liquid slag layer; the slag solidification was not modelled in this work. The cooling of the copper mold was considered by introducing the heat transfer coefficient (HTC) between the mold and water box (excluded from the simulation). The secondary cooling zone below the mold exit was also presented by HTC boundary condition. A thin (3 mm) slag skin was introduced between the mold and solidified shell, playing a heat resistance role like in a real CC process.

### 3.1. Simulation Results

Typical simulation results are presented in Figure 5. Figure 5a displays the temperature distribution in the melt, the slag layer, the CC mold, and the SEN material. Temperatures are scaled between the liquidus and the casting temperature, so it is possible to detect areas

where solidification appeared. Figure 5b shows the velocity magnitude distribution and the direction of the melt flow. The modeled convection inside the viscous slag layer can be clearly observed. This is an important achievement of the present study, since only limited number of publications have simultaneously presented the heat transfer and convection in the melt and liquid slag.

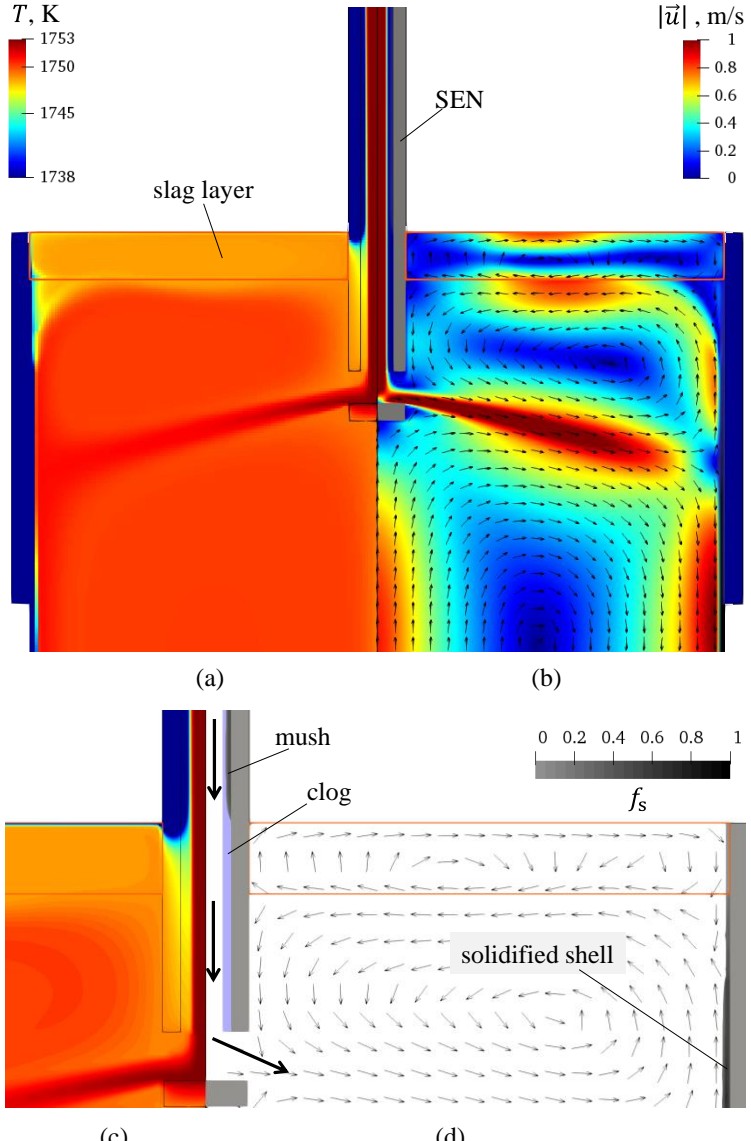

(a)　　　　　　　　(b)

(c)　　　　　　　　(d)

**Figure 5.** Simulation results overview: (**a**) temperature field; (**b**) velocity magnitude distribution; (**c**) local temperature in the SEN/slag region; (**d**) clogging and melt solidification inside SEN.

One can see a typical double pattern in the mold pool. According to the developed coupling algorithm (see Section 2.4 and Figure 3), the momentum of the upper roll was transferred at the programmed interface to the slag phase, causing a reverse flow inside the slag layer (Figure 5b).

Convective-dominant heat transfer can be observed, thus suggesting that the recirculation of the slag layer had a direct impact on the temperature distribution (Figure 5a). The slag (cooled through the refractory SEN, the hot copper mold side, and the air interface) was continuously mixed with hot bottom layers, heated up by the fresh melt, and fed through the SEN ports.

Due to clogging or low SEN insulation, a temperature pattern can dramatically changes (Figure 5c). In reality, this can cause slag solidification in this area, which is highly undesired [50].

Our considered scenarios included the possible formation of the parasitic solidification with and without the formed 12 mm clogging layer (Figure 5d). A detailed study of the simulation results is presented in the next section.

### 3.2. Results Analysis and Discussion

Initially, the phenomenon inside the SEN was studied (Figures 6 and 7) since it had a direct impact on the melt flow and superheat distribution inside the CC mold. The SEN profiles were scaled down by three times in the casting direction to make the solidification profiles and clog layer more visible.

The top figures represent the final state of the SEN channel for the different simulated scenarios. The presence of clogging or parasitic solidification is indicated. The bottom chart shows the temperature distribution along the SEN inner wall (line A-A) and the corresponding height of the mush. The left axis refers to temperature, and the right axis refers to mush height.

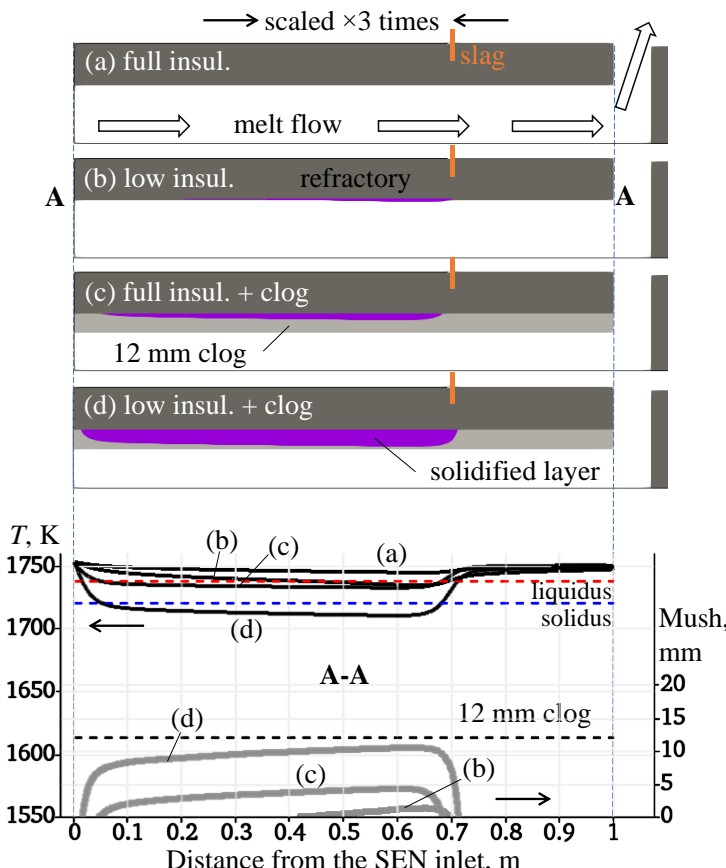

**Figure 6.** Solidified steel inside the SEN (top figures (**a**)–(**d**)) and the temperature distribution/mush thickness along the SEN inner wall (line A-A): (**a**) insulation of the SEN with a fiber layer; (**b**) low insulation of the SEN; (**c**) full insulation of the SEN and formation of the 12 mm clog layer; (**d**) low insulation of the SEN and formation of the 12 mm clog layer.

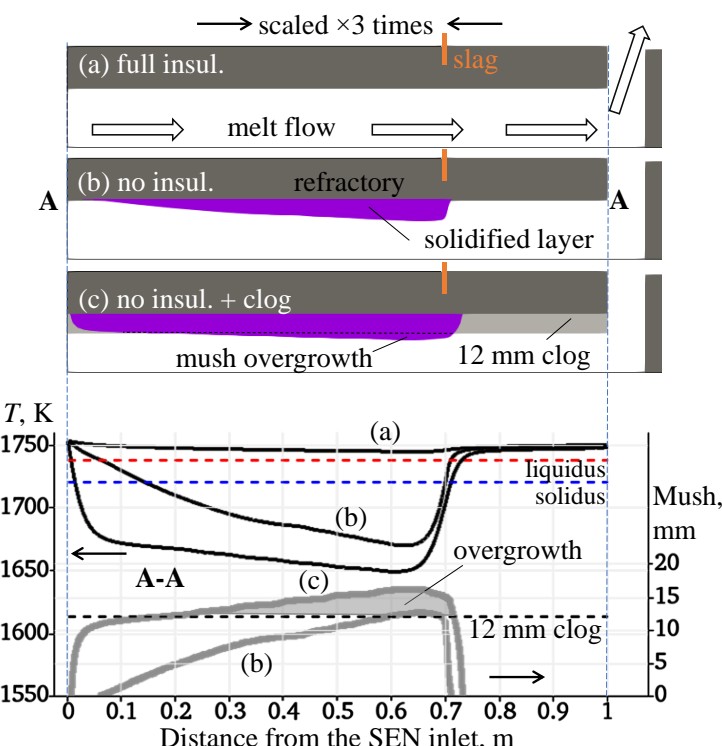

**Figure 7.** Solidified steel inside the SEN (top figures (**a**)–(**c**)) and the temperature distribution/mush thickness along the SEN inner wall (line A-A): (**a**) insulation of the SEN with a fiber layer; (**b**) uninsulated SEN without the 12 mm clog layer; and (**c**) uninsulated SEN with the 12 mm clog layer.

As can be seen in Figure 6a, for the SEN insulated with an outside fiber layer, the inner channel was clean from parasitic solidification (top figure) and the melt temperature along the refractory wall (bottom chart) was significantly higher than the liquidus temperature. Moreover, it only dropped by several degrees in the coldest region. When the thermal resistance of the fiber insulation was lowered by three times (Figure 6b), the inner SEN temperature dropped below liquidus temperature, and a thin solid steel layer was formed. This case reflects the possibility of the operational wear of the insulation layer or the initially insignificant thickness of the fiber layer.

Next, both cases were simultaneously investigated with the formation of the 12 mm clog layer. For the case with the insulated SEN and the 12 mm clog layer (Figure 6c), the melt started to solidify inside the porous clog since the hot melt could not provide enough superheat. As such, the presence of the clog promotes parasitic solidification, even inside a properly insulated SEN.

As expected, the reduction of the SEN insulation led to the stronger solidification of the entrapped melt inside the clogging material (Figure 6d). The height of the mush was comparable with the clog thickness.

For the uninsulated SEN (Figure 7), all previously observed phenomena were more pronounced and a thick solidified layer was formed (Figure 7b). When clogging was additionally considered, the parasitic solidification could even overgrow the clog height (Figure 7c). Of course, these conclusions arise from numerical analysis, but the physical trends are clear.

Changes in the flow were detected between different simulation scenarios. The original flow pattern is presented in Figure 8a, with a double roll in the mold cavity and an opposite recirculation zone inside the slag. When the formation of the clog layer was considered, the flow in the mold was accelerated because an effective diameter of the SEN was reduced for the same throughput (Figure 8b).

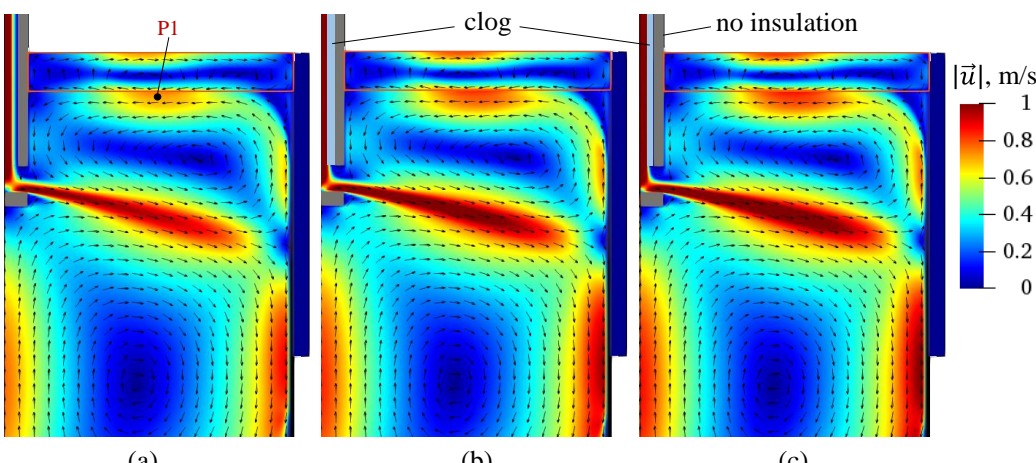

**Figure 8.** Simulation results for the mold flow: (**a**) initial setup with the fiber insulation of the SEN; (**b**) formation of the 12 mm clog layer; and (**c**) uninsulated SEN with the solidified layer overgrowing the clog (see Figure 7c).

When the uninsulated SEN was modelled, the parasitic solidification layer overgrew the clogging (see Figure 7c) and the jet flow was further accelerated (Figure 8c). Due to clogging and parasitic solidification (Figure 8b,c), intensified flow led to the stronger impingement of the hot melt into the solidified shell and destabilized the meniscus flow, thus enhancing the risks of slag entrapment.

The temperature distribution for the initial (ideal) casting conditions is shown in Figure 9a, displaying hot jets and superheat distribution in the mold cavity and the slag layer. The formation of SEN clogging caused an overall temperature drop (Figure 9b), which was further decreased by insulation failure (Figure 9c). Therefore, both phenomena of the clogging and parasitic solidification inhibit superheat transport into the mold and towards the slag layer.

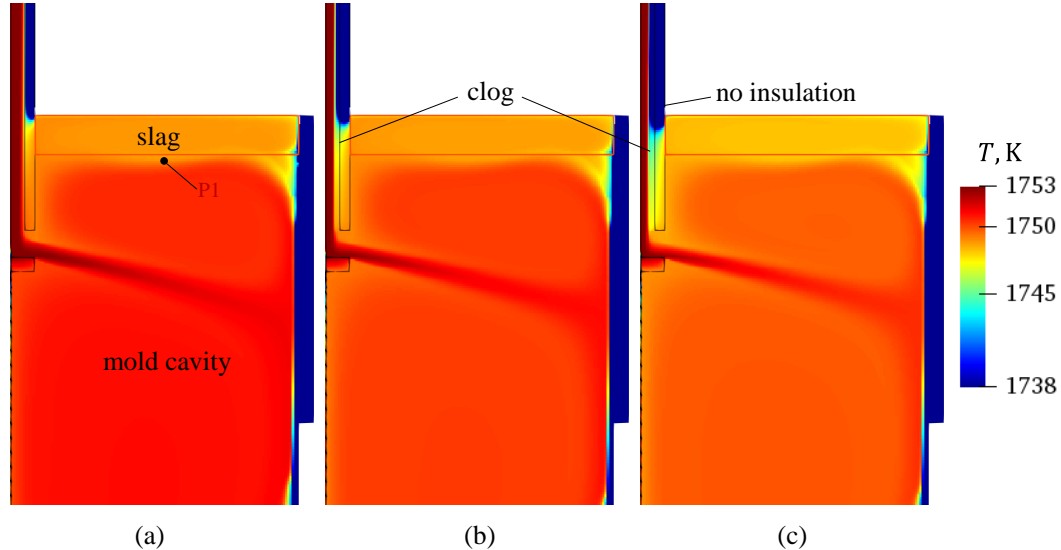

**Figure 9.** Simulation results for the temperature distribution in the mold cavity and slag layer: (**a**) initial setup with the insulated SEN; (**b**) formation of the 12 mm clog layer; and (**c**) uninsulated SEN with the solidified layer overgrowing the clog (see Figure 7c).

As observed from the simulation results (Figures 6, 7 and 9), the inclusion of the convection in the slag layer enables the more precise description of the heat transfer in the

SEN region, and the parasitic solidification could appear even below a top line of the slag layer.

Details of the flow profile and the temperature distribution inside the melt, clog layer, and refractory wall are shown in Figures 10 and 11. The data collection line B-B is located 50 mm above the slag layer and approximately coincides with the location of the thickest solidified layer. The left axis refers to velocity profiles (in blue shades), while the temperature curves (in red shades) are scaled on the right. Dashed curves indicate the presence of the 12 mm clog considered in the simulation.

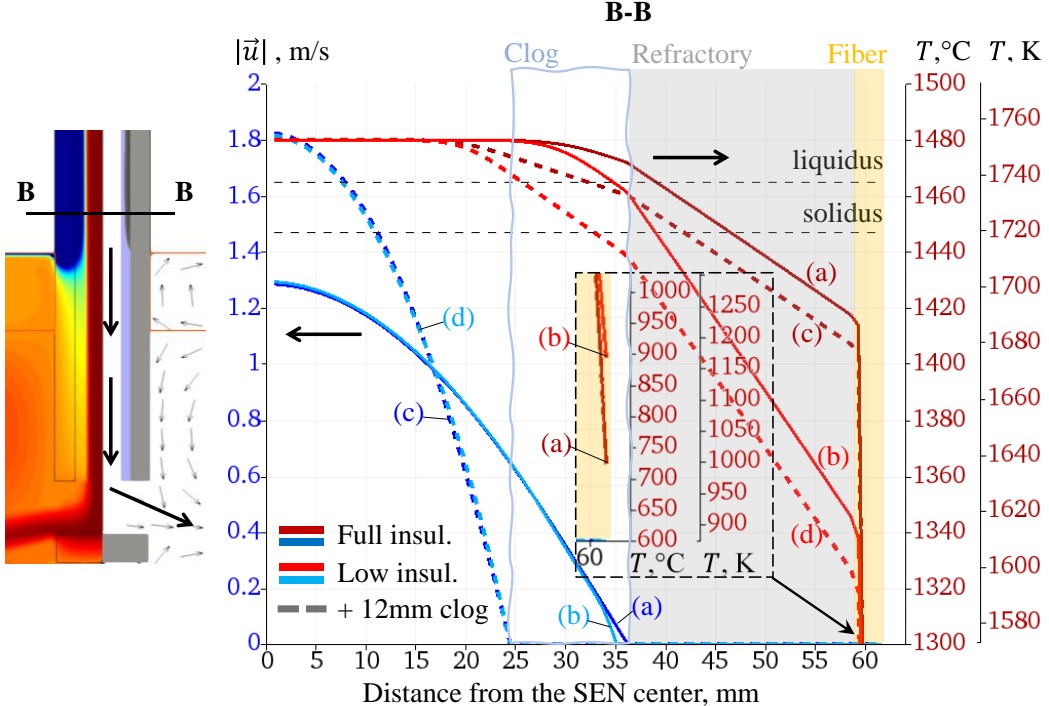

**Figure 10.** Velocity magnitude profile and the temperature distribution along the B-B line 50 mm above the slag layer for the cases (**a,b**) of a fully and partially insulated SEN without clogging (solid lines) and (**c,d**) considering a uniform 12 mm clog layer (dashed lines); the SEN outer surface temperatures are shown in a zoomed view.

Two scales were used for the in degrees Celsius and in Kelvins to make comparisons with referenced measurements [50] more convenient for readers. In a zoomed view for the right bottom corner of the temperature plot in Figures 10 and 11, the SEN outer surface are shown. The scaling was applied to see the solidus/liquidus region in more detail.

For the cases without the clog (blue solid lines (a) and (b) in Figure 10), the flow profiles slightly deviated because a thin layer of the parasitic solidification was formed when the fiber layer resistance was three times lower (see also Figure 6b). Therefore, the boundary layer shifted towards the SEN center; the peak velocity value was slightly higher.

The changes due to damaged SEN insulation were barely noticeable for the melt velocity field; however, they were really pronounced for the temperature profiles (red curves (a) and (b) in Figure 10); the temperatures were generally lower across the refractory and fiber layers. The SEN outer surface temperature increased from the 725 °C in the insulated case to 900 °C when the insulation failed, indicating stronger heat losses (see curves (a) and (b) in the zoomed view in Figure 10a,b). This resulted in a temperature drop in the boundary layer along the inner refractory wall below liquidus and initiated steel solidification.

The simulated surface temperature for the insulated SEN (Figure 10, zoomed view, curve (a)) coincided well with the plant trials from RHI Magnesita GmbH [50], where the measured fluctuating temperature history had an average value of 725 °C.

No difference can be observed in the velocity profile for the clogged cases (dashed blue lines (c) and (d) in Figure 10) with the full and low SEN insulation because the flow stopped at the border of the clog region. However, the temperature profiles significantly deviated due to the higher heat losses at the outer SEN surface, which led to stronger solidification inside the mush, as was previously shown in Figure 6c,d.

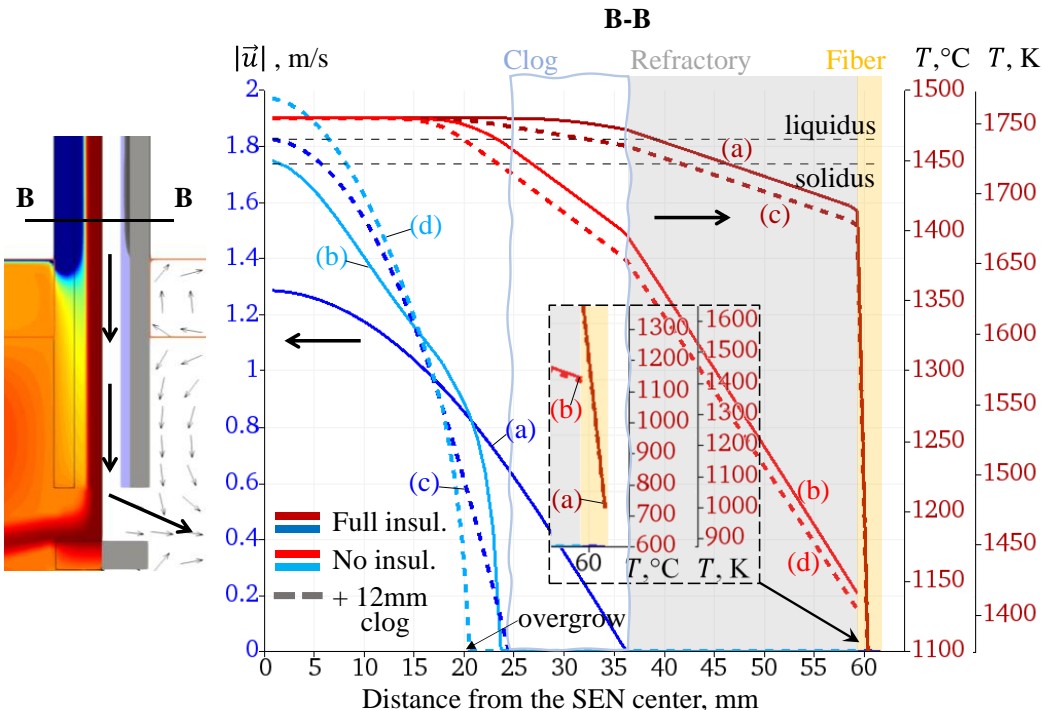

**Figure 11.** Velocity magnitude profile and the temperature distribution along the B-B line 50 mm above the slag layer for the cases (**a**,**b**) of a fully and uninsulated SEN without clogging (solid lines) and (**c**,**d**) considering a uniform 12 mm clog layer (dashed lines); the SEN outer surface temperatures are shown in a zoomed view.

Figure 11 shows a comparison of the "ideal" insulated case against an extreme case with an uninsulated SEN. This scenario is unacceptable for CC operation; however, numerical modeling provided us a tool to consider it for the sake of consistency.

Due to the strong heat losses, a thick solidified steel layer was formed inside the uninsulated SEN (Figure 7b), which reduced the effective inner radius. Thus, the flow accelerated to compensate for the melt throughput (light blue solid line (b) in Figure 11). The outer temperature rose from 725° by more than 400 °C (curves (a) and (b) in the zoomed view in Figure 11).

In the case of the uninsulated SEN with clogging, the parasitic solidification overgrew the clog layer (see Figure 7c and the marking in Figure 11) and caused flow acceleration by up-to 2 m/s (curve (d) in Figure 11). It must be noted that in the present study, clog growth was not modelled. As such, it is highly probable one can expect further clog deposition over the mush region.

The simulation results evidence that clogging promotes solidification, even with a properly insulated SEN, and it significantly enhances clogging if the fiber insulation is damaged. Moreover, the clogged region is "supported" by the solidified steel skeleton and can further develop SEN blockage [63,64].

## 4. Conclusions

The general influence of SEN clogging and parasitic solidification on mold and slag flow, as well as on superheat transport, is presented and discussed here. It was found that the melt flow in the mold was accelerated and destabilized; the superheat dropped due to the clogging and parasitic solidification inside an SEN bore.

The melt velocity profiles and temperature distribution inside an SEN were detailed and analyzed with the aim to reveal the conditions of the parasitic solidification.

The origin of SEN clogging is an important topic for all casting producers. The current study followed previous work and conducted experiments [50,63,64] to provide insights, showing that in conventionally produced SENs, one of the dominant reasons for the blockage and superheat losses is the coupled phenomena of clogging and parasitic solidification promoting each other.

It was shown that the proper insulation of an SEN;s outer surface is important. Damage of the fiber layer can cause a parasitic solidification of the melt adjacent to the inner refractory wall. As proven by the measurements and observations done by RHI Magnesita GmbH, it is required to not only provide an SEN wall with low roughness but also to apply special insulation and anticlog linings that assist in reducing heat flux and energy losses through the SEN material [50].

Our numerical results were verified by comparison with the pyrometric temperature measurements by RHI Magnesita GmbH during the real CC process [50], as well as by the observation of a worn-out SEN.

In Figure 12, a quantitative comparison of clogging and parasitic solidification on the mold sub-meniscus flow and superheat is presented. The results are given for monitor point P1 (see Figures 8 and 9), located in the mold center just below the melt/slag interface. It can be observed that the sub-meniscus velocity grew and the superheat dropped due to the combined effect of SEN blockage and heat losses due to the clogging and formation of the parasitic solidification.

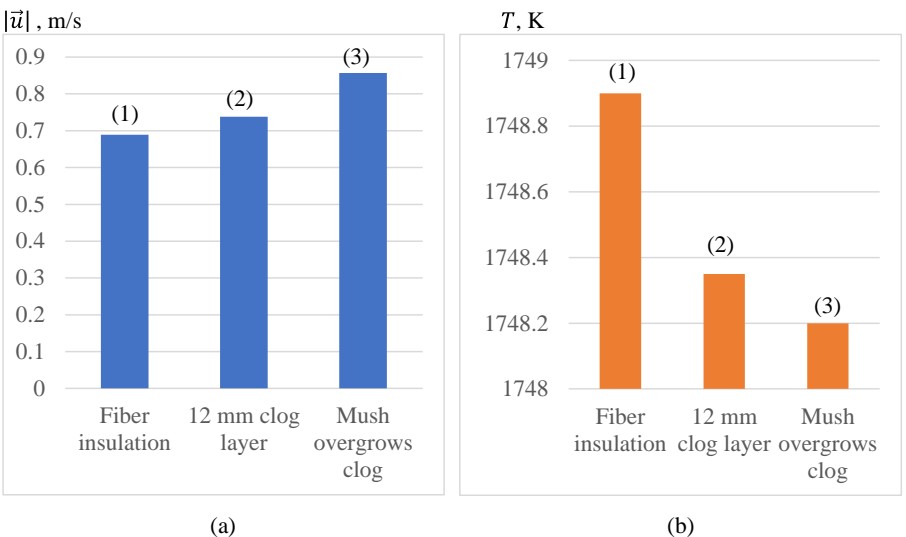

**Figure 12.** Sub-meniscus velocity (**a**) and the temperature change (**b**) at monitor point P1 marked in Figures 8 and 9 for the corresponding cases: (**1**) initial setup with the insulated SEN; (**2**) formation of the 12 mm clog layer; and (**3**) uninsulated SEN with the solidified layer overgrowing the clog (see Figure 7c).

The presented work operated with real casting conditions, in contrast to previous studies connected to experimental parameters and high melt velocities [63,64]. Therefore, it allowed us to detect that parasitic solidification appeared due to the partial failure of SEN insulation.

The current study presents a novel single mesh approach, which is robust and computationally efficient, to simulate complex multi-phase phenomena during the CC process. A new method to couple the momentum transfer at the liquid interface was developed and applied. Using the OpenFOAM® finite volume method libraries to design an in-house solver showed the flexibility and power of the open-source framework for industrial simulations.

The present work is the first step to approve a new technique for future 3D simulations, including additional physical models. The developed model suits the wide application range in the continuous casting field. However, the limitations of the suggested approach must be noted.

The assumption of a flat melt/slag interface did not allow us to tackle, e.g., meniscus waving, slag infiltration, or possible entrapment. On the other hand, this model can be directly integrated with a magnetohydrodynamics model to reflect the Lorentz force during the electromagnetic braking, and it can partially account for slag solidification and melting; argon gas injection and the non-metallic inclusions motion can be tracked by a discrete-particle model.

**Author Contributions:** Conceptualization, methodology, software, writing—original draft preparation, A.V. and J.B.; writing—review and editing, A.K., M.W. and A.L.; data and validation, Y.T., G.H., G.N. and J.W. All authors have read and agreed to the published version of the manuscript.

**Funding:** This research received no external funding.

**Institutional Review Board Statement:** Not applicable.

**Informed Consent Statement:** Not applicable.

**Data Availability Statement:** Not applicable.

**Acknowledgments:** The authors acknowledge the financial support by the Austrian Federal Ministry of Economy, Family and Youth and the National Foundation for Research, Technology and Development within the framework of the Christian Doppler Laboratory for Metallurgical Applications of Magnetohydrodynamics.

**Conflicts of Interest:** The authors declare no conflict of interest.

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
