# Peer review of "On Modelling Parasitic Solidification Due to Heat Loss at Submerged Entry Nozzle Region of Continuous Casting Mold"

_metals, doi:10.3390/met11091375_

Round 1

Reviewer 1 Report

Dear Authors,
You presented a well-structured and scientifically well-written manuscript aimed at solving a specific technology, the сontinuous сasting. In your investigation, you elegantly reduced the solution of a three-dimensional task to a one-dimensional one with a sufficient level of justification for the assumptions. The solution of a three-dimensional task, taking into account all the multifactorial nature of the parameters, can lead to a dead end. Since this process describes differential equations with degenerate conditions.
Despite the high level of research organization and  the results discussion.

I have some comments.

In your model, you assume steel and slag to be a two-phase medium. In fact, there is a gas phase.
Also, You don't pay your atention to a chemical interaction of phases.
And most importantly, I would like to see a universal model that simulates not only the process for the system of steel and slag, but also for others ones ... But this is just my wish.

Good luck!

Author Response

Dear Reviewer, thank you very much for your comments! We took them into account to improve the manuscript. Please find our response attached.

Sincerely,

Dr. Alexander Vakhrushev

Reviewer 2 Report

REVIEW

on article

On Modelling Parasitic Solidification Due to Heat Loss at Submerged Entry Nozzle Region of Continuous Casting Mold

Alexander Vakhrushev, Abdellah Kharicha, Menghuai Wu, Andreas Ludwig, Yong Tang, Gernot Hackl, Gerald Nitzl, Josef Watzinger, and Jan Bohacek

SUMMARY.

The article is devoted to the mathematical modeling of the problems associated with the continuous casting of steel. Improvement of the technological process of continuous casting of steel, its optimization to obtain a higher quality product is an important scientific problem. Experimental study of the processes of forming continuous ingots is related to significant difficulties. Therefore, theoretical study plays an increasing role, in particular, in methods of numerical modeling using modern high-performance computers.

Important technological factors that determine the quality of the surface and the central zone of the ingot are the stable casting speed, control of the secondary cooling modes to stabilize the temperature of the ingot surface.

The authors proposed a mathematical model for the numerical analysis of the continuous casting process, based on a modification of the classical equations of fluid flow. Physically, the process takes place in several stages and is a multiphase medium; therefore, the analysis was carried out numerically. Next, the authors focused on the clogging of submerged entry nozzles (SEN) and parasitic solidification, which is logical, since it is associated with the stability and quality of the continuous casting process.

The reference list contains 82 items.

Title, Abstract, and Conclusion correspond to the content of the article.

However, there are shortcomings and ambiguities that need to be corrected.

COMMENTS.

  1. The authors have to redo the Abstract and bring it in line with the requirements of the Metals journal. The scientific novelty is poorly defined. The authors need to highlight it. Editors strongly encourage authors to use the following style of structured abstracts, but without headings: (1) Background: Place the question addressed in a broad context and highlight the purpose of the study; (2) Methods: Describe briefly the main methods or treatments applied; (3) Results: Summarize the article's main findings; and (4) Conclusions: Indicate the main conclusions or interpretations. The abstract should be an objective representation of the article.
  2. Equations 1-4 have a rectangular element, apparently a typo.
  3. The Introduction is too long, maybe split it into subsections and shorten it.
  4. The Introduction is well written, reflects contemporary and relevant articles on the topic. However, I recommend reflecting on the purpose of the study in the concluding part of the Introduction.
  5. It is unclear whether the authors used standard software or developed their own?
  6. Section 3.1. Material properties and case setup is logical to place in section 2 Materials and methods
  7. Figure 11 is hard to read. I recommend numbering the curves and describing them in the text of the article.
  8. In the Discussion section, the authors need to compare the results obtained with those of other researchers. It is desirable to show the limits of applicability of the model.
  9. In the Conclusion section, I recommend quantifying the results.

In general, the article is devoted to an interesting and important scientific problem that will undoubtedly attract the attention of readers.

However, there are many corrections. I recommend the article for publication after a major correction.

Author Response

(The authors gave the same response as above.)

Reviewer 3 Report

General influence of the SEN clogging along with the parasitic solidification on the mold and slag flow as well as on the superheat transport is presented and discussed in this paper.

The melt velocity profiles and the temperature distribution inside the SEN are detailed and analyzed with the aim to reveal the conditions of the parasitic solidification.

It was shown that a proper insulation of the SEN outer surface is very important.

The current study presents a novel single mesh approach, which is robust and computationally efficient, to simulate complex multi-phase phenomena during CC process. A new method to couple the momentum transfer at the liquid interface is developed and applied.

I would like to see some results after the measurements and it’s interesting to compare results of simulations with results of practical measurements.

Conclusions are clearly and are supported by practical measurements but can be improved a little.

English language and style are fine but are necessary minor spell check.

Author Response

(The authors gave the same response as above.)

Round 2

Reviewer 2 Report

All my comments are taken into account and corrected in the text of the article.

I recommend the article for publication